# The Impact of the Church–State Model for an Effective Guarantee of Religious Freedom: A Study of the Peruvian Experience during the COVID-19 Pandemic

Susana Mosquera 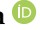

Law Faculty, University of Piura, Piura 20009, Peru; susana.mosquera@udep.edu.pe

**Abstract:** During the COVID-19 pandemic, many governments established important restrictions on religious freedom. Due to a restrictive interpretation of the right to religious freedom, religion was placed in the category of "non-essential activity" and was, therefore, unprotected. Within this framework, this paper tries to offer a reflection on the relevance of the dual nature of religious freedom as an individual and collective right, since the current crisis has made it clear that the individual dimension of religious freedom is vulnerable when the legal model does not offer an adequate institutional guarantee to the collective dimension of religious freedom.

**Keywords:** religious freedom; church–state models; rule of law; religious entities; state of national emergency

## 1. Introduction

It is possible to affirm that basic rights and freedoms have suffered a very direct impact due to the drastic regulations instituted to control the effects of the global COVID-19 pandemic. It was necessary to balance the rights of individuals with those of the community. Freedom of movement, the restriction of the rights of peaceful assembly and manifestation, the adequate provision of the right to education and the effect on the rights to life and health are only some of the essential freedoms stressed by the impacts of this pandemic. However, in this work, the analysis is focused on the restrictions that have affected religious freedom and consequences therein regarding the Peruvian case. The doctrine is unanimous in stating that the established restrictions have effectively challenged religious freedom. However, it is striking that these regulatory restrictions and limitations have not met a greater level of litigation in practice, particularly in Latin America. There were critical voices, but their impact has been limited and, in practice, restrictive measures have been maintained without excessive problems.[1] However, in the United States, we find a rich and varied case law that has even reached the Supreme Court, not once but several times; this court has been able to study the elements of balance between the restrictive regulations and the effective guarantee of the exercise of religious freedom (Madera 2020). In addition, the German Constitutional Court and the French Conseil d'État have had the opportunity to speak on the matter. Additionally, there are many publications of academic work, seminars and articles that analyze the impact of the pandemic on rights and freedoms and further investigations are underway (Opinio Juris 2020).

In February and March 2020, the media coverage of COVID-19 produced images of enormous visual impact: Mecca empty, Pope Francis celebrating Holy Week alone in Saint Peter's Square, the Church of the Holy Sepulchre closed, workers disinfecting the wall of the lamentations in Jerusalem and many other places of worship usually that usually receive a massive influx of people now empty. However, the restrictions soon began to have a distorting and even potentially discriminatory effect (Sarkar 2020). The accusations against those religious groups that decided to maintain their large-scale religious ceremonies were swift in these initial months of the pandemic; soon, the media (particularly, in

New York Times 2020) and science drew attention to those groups, describing them as "super-spreaders" of COVID-19 (Quadri 2020). In this context, governments around the world soon had to approve their protocols to protect health and balance the rights and freedoms of citizens (Sartea 2020). Unfortunately, many of these regulations made a very clear choice: protect health while trying to save the economy,[2] even if that means sacrificing other fundamental rights (especially the right to education or the exercise of freedom of worship).

A perception of arbitrariness thus arises concerning some of the measures adopted by different governments in those first months of the pandemic; there was unjustifiably different treatment towards places of worship. Several human rights organizations indicated that, in the context of the pandemic, many governments used public health policy as an effective restriction of public guarantees and freedoms.[3]

As time passes, this situation has evolved into new scenarios where it is not only civil rights that are affected. It has become clear that it was not only physical health that needed to be considered, because lockdown has had a direct impact on mental health and on spiritual health (Balluerka Lasa et al. 2020). Religion historically plays an essential role in caring for people, and it is evident that this mediating function is carried out through religious functionaries (De León Azcárate 2020). When social distancing measures and the restriction of access to places of worship prevent such spiritual care, a direct effect is produced on the mental health of the faithful because spirituality is a fundamental strategy against pain and illness.[4]

In hindsight, it seems clear that reasonable measures would have offered alternatives to complying with the restrictions established or at least included accommodation for specific cases. Certainly, many countries began to consider this when the first stage of strict lockdown and curfew ended by advancing in phases,[5] marking areas with greater or lesser incidence of the disease and, therefore, accommodating restrictive measures to specific conditions.

The rule of law plays a key role in these contexts; it requires that the decisions of the executive office follow a procedure and are supported by objectifiable elements validated by science to approve measures with a significant social consensus in such a way that they are received and complied with voluntarily.[6] Unfortunately, what we have seen during many of the phases of this pandemic in many countries is that, during the state of emergency, governments have skipped procedures for the correct supervision of their regulatory competences (OECD 2020a). Indeed, governments understood that the emergency makes it necessary to produce quick decisions, but speed—if not accompanied with efficiency and neutral elements that give validity to the decision and social consensus—can rapidly increase the risk of arbitrariness.[7]

Therefore, it seems necessary to reiterate the content that defines religious freedom as a fundamental human right, since it is essential to understand which of its dimensions were impacted by the restrictive measures, considering that it is expressly prohibited to suspend the rights of freedom of thought, conscience and religion during a state of emergency or that of an exceptional nature.[8]

## 2. The Layers of Protection of Religious Freedom

### 2.1. Role of Religion in Society

When liberalism, especially during the French and American revolutionary process, reshaped the political model, it did so by placing the human being as the central axis of the new system, impregnated with a high degree of secularization.[9] In this historical context, it is essential to understand the role that religious struggles played in reshaping the legal system and enforcing the political construction of European states (Gill 2008). Modern states were built in opposition to antiquated structures, and religion was an essential factor in this, among others (such as as culture and language), giving unity to the new national territories (Grzymala-Busse 2020). Thus, it is important to understand the key role that religious freedom had within the category of rights and freedoms during that initial

stage. The role of religious freedom continues to be essential from this political perspective, since it is through freedom of thought that an essential pluralism is guaranteed for the strengthening of democracy and the rule of law.[10]

There is no doubt that religiosity is a multidimensional (Fetzer Institute 1999) construction composed of feelings, thoughts, experiences and behaviors, expressed by the individual but developed through the collective dimension of religious teachings.[11] For such reasons, describing and protecting religious entities is important because they create the basis to specify manifestations of personal faith protected by individual religious freedom. The dual nature of this approach is useful to understanding the complex relationships established on the grounds of the exercise of religious freedom (Yildirim 2017). This relationship can only be explained by the interaction of three different actors: citizen, state and church. The citizen–state relationship can be described under the constitutional parameter of the protection of rights and political freedoms. The faithful–religious entity relationship must be expressed through the free choice of faith made by the act of individual conscience. Finally, closing the circle, there should be the relationship between the religious entity–state established through the institutional[12] status of religious groups.[13]

In this way, religious entities are included in the category of organizations that represent civil society,[14] and following Stepan's approach, they play an important role in strengthening democracy (Stepan 2000). Considering the pandemic as a human and social crisis, the solution must incorporate these two elements: the individual on the one hand and the social element on the other.[15] Undoubtedly, our life in society is built in circles, and religious freedom is related to well-being indicators that allow one to affirm the importance of its effective protection and constitutional recognition (Cross 2015). Although we have indeed advanced towards a secularization that separates political from religious power, we should not be mistaken; while church–state separation occurs at the institutional level, this does not translate precisely into the separation between politics and religion on a personal level. Citizens have an intimate environment of beliefs, whether political, ideological or religious, and these beliefs help build the individual's personal identity (Aldridge 2000).

At this point, it is appropriate to remark upon the particular relationship that is established between the individual and collective dimensions of religious freedom (Scolnicov 2010). Such a dual dimension must be considered because, without beliefs, worship and doctrine, the individual dimension would be empty of content.

*2.2. Content of Religious Freedom*

To affirm that the right to religious freedom has been violated, we must know what this right entitles or, rather, where the impact is located within its radius of action.[16] For the description of this essential right, the text of Article 18 of the Universal Declaration of Human Rights was used.[17] This document defines the classic trinity of the right as: "freedom of thought, conscience and religion". Curiously, almost everyone (agnostics, liberals and conservatives) feels comfortable with this formula, probably due to the lack of precision that we find in the terms used in Article 18.[18]

The literality of Article 18 was used later in other human rights treaties, particularly in the European Convention on Human Rights and in the International Covenant on Civil and Political Rights, so it is relevant to understand the meaning of this right of religious freedom in the Universal Declaration according to the drafters' intended meaning.[19] From a political analysis that can be made regarding Article 18, some interesting conclusions can be drawn (Lindkvist 2013): the controversial rights of religious minorities that caused many problems during the drafting of the Treaty of Versailles, and in the context of World War II, disappear from the text. As a result of this withdrawal of protection of minorities (understood as organized groups), the Universal Declaration concentrates its efforts on protecting the individual dimension of the right to freedom of thought, conscience and religion, giving significant weight to the free expression of conscience,[20] including the controversial right to change religion.[21]

Therefore, considering this lack of precision, we thus perform intellectual reflection to understand why these three rights appear linked in the same article. First, we must understand that freedom of thought, freedom of conscience and religious freedom appear united in their category of subjective rights, as the right to which the person who constructs his or her identity in religious matters is entitled. Thus, understood as a subjective right, its innermost core will always be composed of freedom of thought. This freedom, exercised in a very personal sphere that is guaranteed immunity from coercion, reaches the external sphere through freedom of expression.[22]

Undoubtedly, freedom of thought is not only composed of a religious dimension but also serves as a basis for providing content for the exercise of many other rights (the right to vote, the right of association, the right to demonstrate, and the right to union participation, to name a few of the classic rights that are consolidated under freedom of thought). Once endowed with content, this freedom of thought escapes the inner core of the person and allows the individual's consciousness to be expressed. From that moment on, we say that someone acts conscientiously because it is consistent with their ideas and they would prefer to make a sacrifice before acting against them. However, at this point, we must ask ourselves what gives content to freedom of conscience. Is it a purely individual act or, on the contrary, does it require a doctrine, teaching or ideology, previously constructed by a religious community? It seems that an individual formulation of conscience that is not supported by religious or ideological doctrine has few options to succeed cases where the conscience is protected through conscientious objection.[23] Therefore, although the origin of consciousness can be placed at the forum internum (freedom of thought), the act of consciousness is materialized in the forum externum.

Finally, the third and last element that makes up the trinity contained in Article 18 is religious freedom. This is not the simple ritualization of faith, but the logical path that communicates the internal sphere of freedom of thought that builds the identity of the person in religious matters, which leads them to act in the forum externum in a manner consistent with their faith, and is accompanied by the effective ritual expression of worship in specific religious modes and contexts. Understanding religious freedom only as a protection of worship acts without also recognizing that the inner protection of the freedom of the faithful's conscience and thought is necessary for the essential dignity of the human being would be an incorrect interpretation of Article 18.

Therefore, Article 18 should be understood as protection in levels or layers. From an internal and generic level, the guarantee of the right is non-interference in freedom of thought; its main protection tool is ideological pluralism as an essential guarantee, which helps to build a free democratic model.[24] The next level of content is specified in the outer layer, which is freedom of conscience building upon freedom of thought, and a final and more specific layer is formed by the different ways of externalizing the religious consciousness worship acts of each believer's community.

Therefore, in the innermost layer, individual nature is much stronger (the guarantee of freedom of thought has to be materialized on the individual level of the person), but as we move towards the outer layers of the exercise of religious freedom, we realize that the content of this freedom is built over a collective dimension. Hence, the importance of structuring the exercise of this right dually, since without individual freedom, there is no effective protection for the free exercise of religion, and without the collective dimension, the external sphere would be left without content.

Finally, religious freedom also has a political dimension (not as a human right but as a constitutional principle) that guides the actions of public powers. This dimension obliges the state to assume responsibility to effectively promote and protect individual rights, creating specific conditions (Viladrich 1980; Mosquera 2018). This explanatory description of the content of the right to religious freedom, which allows us to understand its relationship with the principles that guide the model of church–state relations, serves to clarify the understanding used in this work within the discipline of the study of ecclesiastical law.

Thus, if the role of these first freedoms is important, how does one explain why the protection of these freedoms has not been valued during the pandemic? In the trinity, freedom of thought has a close relationship with freedom of expression—thought lives in the inner core and reaches the forum externum layer through freedom of expression. Thus, with such a reference, how does one explain why/how freedom of religion was forgotten so quickly during this pandemic?

*2.3. Suspension of Rights during the COVID-19 Pandemic*

Emergency powers should be used within the parameters provided by international human rights law;[25] such powers should be time bound and only exercised on a temporary basis, with the aim to restore a state of normalcy as soon as possible.

In April 2020, the office of the United Nations High Commissioner for Human Rights made clear the limits that these emergency measures should have. "The restriction must be 'provided by law'. This means that the limitation must be contained in a national law of general application, which is in force at the time the limitation is applied. The law must not be arbitrary or unreasonable, and it must be clear and accessible to the public. The restriction must be necessary for the protection of one of the permissible grounds stated in the ICCPR, which include public health, and must respond to a pressing social need. The restriction must be proportionate to the interest at stake, i.e., it must be appropriate to achieve its protective function; and it must be the least intrusive option among those that might achieve the desired result. No restriction shall discriminate contrary to the provisions of international human rights law. All limitations should be interpreted strictly and in favour of the right at issue. No limitation can be applied in an arbitrary manner. The authorities have the burden of justifying restrictions upon rights" (OHCHR 2020).

Nevertheless, we have the impression that the balancing tools used to assess the standards in the context of the pandemic have been insufficient (Lebret 2020; Sun 2020). It is likely that they have not been respectful of the condition of "essential service" that religion and education have for the human being. As Criddle and Fox-Decent point out, during public emergencies, "states must tailor their responsive measures to minimize the potential impact on human rights" (Criddle and Fox-Decent 2012).

Special concern arises from the lack of reasonableness of the measures applied and the effect that they have had on the essential content of the right to religious freedom. As mentioned, the intense application of measures that restrict fundamental rights by governments around the world makes it necessary to carefully consider which regime, mechanisms and control procedures have declared a suspension of rights in the specific case of freedoms related to the dimensions of a faithful. Article 18 of the Universal Declaration of Human Rights reminds us that religious freedom is a right that cannot be derogated (Bielefeldt 2020). In General Comment No. 22, the Human Rights Committee made it very clear that any type of restriction on the exercise of this fundamental right would have to be carried out by law.[26]

The Human Rights Committee, in its General Comment No. 29, establishes the specific safeguards that states must offer when unilaterally derogating Article 4 of the Covenant. Measures must be of an exceptional and temporary nature, and two fundamental conditions must be met: "the situation must amount to a public emergency which threatens the life of the nation, and the State party must have officially proclaimed a state of emergency".[27] As early as April, the Interamerican Commission of Human Rights called on the OAS States to ensure that the emergency measures that they adopt to address the COVID-19 pandemic were compatible with their international obligations.[28]

Thus, a necessary question arises as to whether the COVID-19 rules have restricted the right to religious freedom or have derogated its effective exercise (Du Plessis 2020). Freedom of worship should be protected by the freedom of religion (as the third dimension of the right protected by Article 18, UDHR). Such freedom of worship includes the celebration of sacraments, funeral rites, and spiritual assistance to people deprived of liberty (Ramírez Navalón 2020). Since these are the most frequent manifestations (a non-closed

list), it is important to note that not all religious groups can adapt to a virtual format as required in the context of the pandemic. Therefore, the first situation that arises is related to inequality among denominations because not all worship can be transferred online, thus having a direct consequence on the faithful's rights. Many confessions have had to adapt their acts of worship to virtual celebrations, which are not always in the same conditions. Many small communities do not have enough resources to follow religious ceremonies in this virtual mode. Rural populations, native communities and religious minorities that do not have an official channel of communication with government authorities have seen the exercise of their religious freedom suspended as they have been unable to access a valid way to worship. In all these cases, the restrictive regulations on religious freedom during the pandemic have served to expose the social inequality suffered by these groups. For other communities, the difficulty of adapting to the virtual format is not due to economic, logistical or capacity reasons, but is due to the essential theology and the practice of the sacraments in that religion.[29]

In the specific case of religious freedom, although some layers of content may indeed maintain a minimum guarantee when its exercise is transferred to a remote modality and without the presence of a minister,[30] it is undoubted that the effective realization of liturgical acts that involve a collective sacramentality and that require the presence of the faithful and the minister are impossible to transfer to the virtual world.[31] The previously mentioned "religiosity test" implies the support that the collective religious structure gives to the individual worship. Without a collective dimension, without a religious entity, the individual dimension loses its content. Therefore, an excessive postponement of collective worship acts can ultimately affect the very existence of religious freedom.

Considering pandemic circumstances where regulations have limited religious freedom by categorizing worship a non-essential service, it is worth questioning the importance that the model of church–state relations has for the effective protection of the right to religious freedom. It cannot be forgotten that the health or education sectors, among others, have been the natural field of work for religious entities, especially in the Christian world. Nevertheless, in a context where collaboration would have been decisive (to manage humanitarian aid and health and educational services), the state prefers to postpone collaboration with entities. This forces us to reflect on the proper functioning of the "balancing" tools in the exercise of fundamental rights. Does the lawmaker, the executive or the judiciary play an adequate role? Should the incorporation of other actors be considered? Let us not forget that church and state relationships are tripartite relations: the faithful, the state and religious entities.

Therefore, it is possible to suggest that in the context of this pandemic, the state has resolved the situation of the faithful, not through a direct state–citizen relationship nor through the institutional state–religious entities relationship, but through a state–faithful relationship, which, as a "non-confessional" or neutral state, is expressly disqualified.[32] Considering that the citizen has more difficulties in this bilateral relationship to claim non-compliance by the state in the effective protection of the rights and freedoms linked to religion and worship, it seems that this was a task that religious entities had to assume: to defend the essential content of the religious dimension of the individual.

## 3. The Church–State Model and Effective Protection of Religious Freedom

After analyzing the generic framework of this work and considering the essential content that we must assign to religious freedom as a fundamental right in its individual and collective dimensions, we must also consider a matrix right that manages to manifest all these layers of content through an exercise combined with other rights and freedoms, such as freedom of assembly, association and demonstration. This peculiar circumstance makes the discipline of ecclesiastical law adopt an interdisciplinary approach since the object of study "religion" displays its effects through different branches of the legal system (Sandberg 2008). This interdisciplinary nature explains why we can find works on the role of religion in society from different approaches that include constitutional law, political

science and the sociology of religions, to name a few. It serves as a clarification that this work is based on the principles and models that are studied from the perspective of civil ecclesiastical law.

### 3.1. The Church and State Systems

It was believed that the effective and full protection of religious freedom could be achieved through the incorporation of that right into constitutional texts, completing internal protection with a network of human rights treaties. One would think that as long as there was constitutional recognition of the protection of religious freedom, we would have achieved sufficient individual guarantee for the exercise of this right. However, as far as the individual guarantee of the exercise of religious freedom is intrinsically related to the institutional position that a religious group has within the legal model, it is possible to affirm that when there is no protection for the collective dimension of religious freedom, the natural consequence is that the individual exercise of the acts of worship of that faith is severely affected or restricted (Nieuwenhuis 2012).

The literature tends to classify church–state models into three large groups: the state–church model, the secular separation model and the collaboration or hybrid model (Ferrari 2013; Robbers 2019; Madeley 2015). Therefore, we can roughly consider the existence of confessional models (whether real or just sociological confessionalism), models of separation (more or less neutral before the role of religion in society), and models in the cooperation category. It is possible to affirm that the differences between these models are conditioned by historical, cultural and social factors, which, in practice, generate a different combination of alternatives between apparently sibling legal models (which sometimes have a root with a similar legal origin) but which present particular sociocultural religious structures.[33] Taking the three basic models (state–church system, separation and cooperationist), and considering Halmai's approach that the model of state–religion relations could determine the state of religious freedom of a given country,[34] we pose the question as to whether some models of church–state relations are better than others.

The experience left by restrictions on human rights during the COVID-19 pandemic allows for investigations to continue for a long time and offers the basis for new decisions that allow for a better approach to these guarantees of freedom in the future.[35] For the moment, what 2020 has left has been an excess of regulatory intervention by governments sacrificing the exercise of public freedoms under the premise of protecting public health. Certainly, not all governments have applied equally restrictive measures and not all of them have opted for radical lockdowns, and when they have done so, not all legal models have acted the same in response to these restrictions on freedoms.

### 3.2. The Distinctiveness of the Latin American Context

Two major subjects of content within civil ecclesiastical law—the protection of the right of religious freedom of the person and the church–state relationships—are the obvious consequence derived from the dual nature (individual and collective) of the right to religious freedom. In this dual plane, it is useful to ask which one benefits the most from the legislative development of the right to religious freedom. Cooperation between political and religious powers is necessary and cannot accept violations of the constitutional principles that define the content of ecclesiastical law as an academic discipline: religious freedom, non-discrimination on the grounds of religion or collaboration and independence and autonomy between the state and religious entities.

In the specific context of South America, it is notable that there has been little litigation regarding the protection of religious freedom during this pandemic context. Only a few cases reached the courts of Concepcion in Chile with lawsuits filed by a group of lay Catholics, by a public worker and one filed by a group of pastors from the Bio-Bio Region. All of them were declared unfounded (Bustamante and Astaburuaga 2020). Another lawsuit filed by a Catholic Colombian lawyer against the Government was declared inadmissible and without grounds.[36] The majority opinion of academics in Latin America is that

religious freedom has been subjected to "disproportionate and unjustified" restriction measures,[37] and that religious freedom was, in fact, suspended.[38]

Perhaps to understand this statement we should point out that the church–state relationship in Latin America is based on relatively young constitutional systems that include a bill of fundamental rights and constitutional clauses of interpretation under human rights treaties and a formal separation (or non-establishment clause) of the church–state relationship. However, in Latin America, the secularization of society is still very low, and the political role of religious entities (especially the Catholic Church) is still relevant (Gill 1998), maintaining some of the historic "regalista" of state–church models established during the 18th century (De La Hera 1992).

The right to religious freedom in Latin America since the end of the 20th century has presented a series of characteristics that need highlighting to be able to appreciate the notable peculiarities more clearly. In the 1980s and 1990s, most countries in the region began constituent processes to overcome dictatorships and military governments. Consistently, the new constitutions recognize the right to religious freedom enunciated mainly as "freedom of conscience and religion", likely due to the influence that the drafting of the American Convention on Human Rights had on the constitutional processes of the region.[39] Along with this constitutional and conventional recognition of religious freedom, it is worth noting another peculiarity of the countries of the region in that one can find the survival of models of formal Catholic and sociological confessionalism with greater intensity. Certainly, this is a new confessional formula compatible with the recognition of religious freedom, but this situation is striking when compared to the evolution of secularization in other parts of the world.

Low-intensity concordant regimes[40] coexist in the region with other relationship systems that are evolving towards a formula of cooperation, with legal registration for different cults. Therefore, in their relationship with religious entities, almost all the countries of South America (except for Mexico and Uruguay as models of separatist secularism) are considered collaboration or hybrid church–state models.[41] The Catholic Church is the sociological majority denomination, followed by other Christian denominations making up the main minority groups; Muslims, Jews and other religious groups make up a religious minority in Latin America. Many countries have a registration system for religious entities, designed to grant legal personality to religious entities.[42] A logical consequence of the registration includes the possibility of formalizing church–state collaboration agreements.[43] However, to date, such agreements have been difficult to carry out in addition to the lack of interest in establishing them. The sociological majority presence of a group may cancel out the possibility of promoting a true model of collaboration based on religious pluralism since this pluralism is not yet visible in society (Deiros 1991). Perhaps Buckley's suggestion that "the institutional logic of religion-state relations is quite different in consolidated democracies than less competitive regimes" can be applied to this context (Buckley 2018).

Thus, just as it is possible to find exceptionality in religious matters in Europe (Davie 2002), there is also a specific situation in Latin America. Here, two circumstances combine their effects: democracies that are too young (with a high level of corruption and disorganization) and a low level of secularization in society. The difficulty in promoting pluralism that allows for democratic strengthening may, then, have its origin in the role that religion plays in these countries.[44] Benevolent secularism empowering accommodations could be the option to enforce institutional cooperation with the religious entities (Buckley 2015).

### 3.3. Peruvian Church and State Relationship

In Peru, the church–state relationship model has evolved from the Catholic confessionalism of the historical constitutions that excluded any other religion, passing through a model of religious tolerance to finally becoming a model of positive cooperation with denominations that fully recognize and protect the right to freedom of conscience and religion of all individuals and groups (Stanger 1927). This cooperationist model took shape

in 1980 with the signing of an agreement between the Holy See and the Peruvian State to regulate common matters in what turned out to be adequate compliance with Article 50 of the Constitution. However, the development and compliance of the second paragraph of that same article were pending, where it was stated that: "The State respects other confessions and may establish forms of collaboration with them".

To implement these "forms of collaboration", in 2001, a series of changes were initiated within the Ministry of Justice. The internal structure within the Ministry was modified; new functions were assigned to the National Direction of Justice; and the Directorate of Interconfessional Affairs was created with the function of coordinating and promoting the relations of the Executive Power with confessions other than the Catholic Church, as established by the State for the strengthening of religious freedom.[45] It appeared that the collaborative model was beginning to lay its foundations, but its evolution has been chaotic (Santos Loyola 2019).

In line with the new role assigned to the Directorate of Interconfessional Affairs, it was imperative to know with which religious entities or groups the new directorate should communicate, hence the need to establish a control or registration system for non-Catholic religious entities.[46]

A Peruvian lawmaker began to build his church–state model by cataloguing and registering the religious entities that operated in the territory, but did so without considering the consequences that such a process would have to comply with, i.e., the constitutional principle of collaboration contained in Article 50 of the Constitution.[47] The regulation was completed with the Law 29635 concerning religious freedom, where we found the practical formulation of how the implementation of this new way of understanding the relations between the state and the confessions was to be carried out. Collaboration agreements with entities that had established roots in Peruvian society were the axis of this legislative development in which great expectations had been placed (Mosquera 2019).

As a conclusion regarding the evolution of the model, it can be said that there has been no significant progress in specifying the constitutional principle of cooperation between the state and religious entities (not in Peru or any other South American country). In such a way, Peru is a state that formally adopts a constitutional model of cooperation between the state and religious entities, but has not yet come to specify the application of that model with the religious minorities and maintains practices of the confessional model (Mosquera 2020).

Peru in particular, and the region in general, lacks an organized structure of churches that can maintain a firm position in the face of excesses of political power. The weakness of religious entities in their institutional presence has a significant impact on the protection of the individual exercise of religious freedom. During the pandemic, it was evident that the ecclesiastical authorities joined the political authorities, accepting the restrictive norms on freedom of worship dictated by the government. Countries as little secularized as Peru keep the "official" church within a role of "quasi" public structure similar to the one it had at the time of the Viceroyalty. In this sense, the Peruvian model of church–state relations claims to be a model of cooperation, but in practice there is evidence of a disguised confessional model in which the religious factor is not sufficiently independent or autonomous from the state. There is no progress in collaboration with the other religions present in the territory, and without cooperation effective separation is not achieved nor is the pluralism that fosters higher levels of democratic quality.

During the current pandemic context, religious entities have played an essential role in democratic regimes, protesting against restrictive regulations and monitoring governments decisions for their potential restriction on the exercise of the fundamental right of freedom of religion.[48] However, in those models with a low democratic level where religious pluralism is not entrenched in society through an effective mechanism to enforce pluralism and the institutional role of religion in society, we observe models with poor protections for the right to religious freedom (Fox and Tabory 2008). Having an organized structure of religious entities, as with the structure of civil society, is an essential instrument to guarantee the rule of law and promote human rights.[49] If we lack strong institutions that

allow for a balance between the functions of the judicial legislative executive, unity with the functions of the market structures and the organizations that represent civil society, the temptation of an unlimited exercise of power by any of these sectors represents a high risk.

Comparing the responses of different congregations and religious groups regarding the governmental regulations, we could say that their reactions have oscillated between an important pragmatism[50] and a second line of action where they have implemented transformative practices in the exercise of worship;[51] finally, there has been a significant group of religious entities that have opted for the format of resistance and defense of the religious freedom of the faithful.[52] The key to facing one or another type of reaction by ecclesiastical authorities appears to be related to the effective separation that exists between the state and religious entities and also with the institutional guarantee of the collective dimension of religious freedom.

For this reason, applying the first parameter, we find that religious entities have been able to provide a better critical response to government restrictions in those legal models with a better understanding of church and state "twin toleration" (such as France or the United States); and in others, with a high recognition of the collective dimension of religious freedom (such as the models of established churches or the models of collaboration, such as in Greece or Germany). On the other hand, in the context of Latin America, where secularization and the organized institutional strength of religious entities are low, critical capacity has been significantly diminished, and subsequently the individual dimension of religious freedom has been affected.

In the Peruvian case, it is worth asking whether the government conducted any consultations with the leaders of different denominations before implementing the restrictive measures on the freedom of worship.[53] Everything seems to indicate that they did not, and when the Peruvian Episcopal Conference proposed a protocol for religious worship in times of pandemic in May 2020,[54] they likely believed that they would be able to decide when to reopen the churches; however, it was not possible to do so until November, when the government expressly authorized it. This means that in the specific case of Peru, the churches were closed for worship from 15 March to 15 November, when in other parts of the world with similar figures regarding the impact of the pandemic, number of deaths and similar sociological data, they had reopened the churches from June 2020.[55] It is striking that the decision of the Peruvian government to reopen the churches came after two very significant legal episodes: a bill presented on 13 October 2020 to force the reopening of the churches[56] and the publication of guides and directives of the Inter-American System for the protection of human rights in October 2020, very much in line with protecting the religious freedom of the faithful in the context of Latin America. In other words, ultimately, it was external pressure that forced the Peruvian government to relax the restrictive measures on freedom of worship.[57]

The Easter 2021 celebration finally set a good example for comparison. Several regions of Peru entered the maximum risk zone on 29 March, and greater restrictions were established. In practice, this has meant reducing the capacity of places of worship to 0, while restaurants and hairdressers kept their capacity at 40%. Specifically, from Holy Thursday to Easter Sunday, a strict home lockdown regime was established. In comparison, during the same period, Italy entered a strict three-day lockdown to prevent a surge in COVID-19 cases over Easter. Non-essential movement was banned, but people were allowed to share an Easter meal at home with two other adults. Churches remained open, but worshippers were told to attend services within their own regions. All non-essential shops were closed, and cafes and restaurants were running a takeaway-only service.[58]

## 4. Final Remarks

At this point, it seems that it is possible to conclude that the legal tools that allow for balance in the exercise of fundamental rights have not been entirely successful in 2020. The vast majority of governments have decided to attempt to protect citizens' health against the threat of COVID-19, although this has meant sacrificing the exercise of other rights,

including the right to health itself since all other non-COVID-19 diseases have seen their treatments suspended throughout this period as well as other dimensions of health related to mental health and, especially, spiritual health. Together with the right to education (especially concerning the effective protection of inclusive education)[59], the right that has been most infringed upon during the pandemic has likely been freedom of worship.[60]

Political models with a higher level of institutional democracy are those that have offered more guarantees for the protection of public liberties during the pandemic. For this reason, it is possible to remark that models with a weak institutional presence of religious entities have faced a shortage in the protection of individual freedom. Considering the guarantee that religious freedom must be received as a human right, it is especially worrying that religious confessions as structures of civil society suffer such institutional deficits.

Accommodation techniques to resolve these tense situations are possible. Many religious practices have been able to adapt and accommodate to the virtual format (especially in the first stage where scientific information on the modes of contagion of the virus was insufficient). However, as the months progressed, it became necessary to review the legislative measures adopted (since any restrictive measure must be temporary) and, in light of this review, make effective accommodations that would not sacrifice the exercise of rights. Liturgical celebrations in the open air, churches/temples open for individual religious activity with social distancing in place and control measures within places of worship assuming that the faithful can adhere to the established public health measures are just some of the examples of the accommodation measures that could have been taken without having to sacrifice the rights and freedoms of citizens. In this initial balance between individual freedoms and the preservation of the right to health, not all the elements of judgment were taken into consideration for a reasonable assessment of the elements of the debate, and a hierarchical application of rights was made.

In the specific case of the exercise of religious freedom, it is evident that the constitutional guarantee of the individual right to religious freedom requires the institutional complement of protection of the collective dimension of religious freedom. It has become clear that the constitutional framework was insufficient in Peru and that the restriction of fundamental rights did not follow the controls established by international obligations. If the government wishes to maintain some type of restriction on meetings, it must do so through measures that are sufficiently neutral but that at the same time include in the arguments those that refer to the condition of religious freedom as a human right.

The non-discrimination requirements mean that those measures that restrict religious activities in a manifestly different way from other activities with which they retain a relationship and context must have a reinforced justification clause. In other words, religious entities have earned "justifiably different" treatment because their operation is the basis for the effective realization of the right to religious freedom.

The government must be able to convince citizens that there is an interest that justifies the establishment of this limitation on the exercise of rights, and to arrive at this justification, it must apply a careful balancing technique. Not only is it essential that the state demonstrate the existence of a justification for establishing such a limitation but it must also prove that there is no other way to protect that interest. Legal tools help us apply that right. Guiding principles such as the rule of law and the principle of non-discrimination allow the legislator to have objective criteria for making political decisions.

The specific situation in Peru has made evident the precarious development of fundamental rights. It is clear that a deficient guarantee of the right to health, combined with precarious infrastructure, personnel deficits and scarce material resources to deal with a pandemic, has effectively forced the sacrifice of services considered "non-essential" in order to avoid the collapse of the health system; however, some of those non-essential services were as fundamental as a human right. If religious entities do not have a structure that is sufficiently autonomous and independent from the political powers and that can sue against government regulations that restrict the freedom of worship of the faithful, this

layer of individual exercise is hindered because of the scarce guarantee of the collective dimension of religious freedom.

**5. Conclusions**

Basic rights and freedoms have suffered a very direct impact due to the drastic regulations instituted to control the effects of the global COVID-19 pandemic in Peru. In this context, the decision to limit the exercise of freedom of worship has been a response taken by governments, in Peru in particular and in America in general, but has not been directly challenged by the courts. This situation leads us to affirm that the fundamental right to religious freedom has, in fact, been restricted. The opportunity to use reasonable accommodation on the grounds of religion as a broad legal tool was not taken, thereby evidencing the limitations of law in such complex contexts.

The complexity in the particular case of religious freedom is explained by the layers of content that make up this right. We must consider that freedom of thought, conscience and religion should be understood as a right with protection in levels. The guarantee of freedom of thought plays its main role at the individual level (forum internum), but as we move towards the outer layers of the exercise of religious freedom (conscience and religion), the content of this freedom needs to be sustained over its collective dimension. This is the reason that explains the particular relationship between the individual and collective dimensions of religious freedom. However, during the context of this pandemic, the government has resolved the situation of the faithful neither through a direct state–citizen relationship nor through its institutional state–religious entities relationship, but through a state–faithful relationship, which, as a "non-confessional" state, is expressly disqualified.

Consistent with the relationship between the individual and collective dimensions, the individual guarantee of the exercise of religious freedom is intrinsically related to the effective protection that the religious group has within the legal model, so it is possible to affirm that, when there is no protection for the collective dimension of religious freedom, the natural consequence is that the individual exercise of worship is severely affected or restricted. This is the case in Latin America, where the unique understanding of the church–state relation model with a limited guarantee of the collective dimension of the right to religious freedom may explain the limited reaction to the government measures during this pandemic. This allows for the conclusion that church and state policies have direct impacts on the effective protection of the right to religious freedom in its individual dimension.

**Funding:** This research received no external funding.

**Institutional Review Board Statement:** Not applicable.

**Informed Consent Statement:** Not applicable.

**Data Availability Statement:** Not applicable.

**Conflicts of Interest:** The author declares no conflict of interest.

**Notes**

1  During the first months of 2021, Peru resumed strict restrictions that limit the celebration of liturgical acts to the maximum.
2  "The lack of effective response from a number of governments to protect people's health through proven measures such as social distancing and quarantines to flatten the curve of the pandemic is also very concerning. Arguing that the cure would be worse than the disease, some governments have opposed these measures to avoid an economic slowdown" (Bohoslawsky 2020).
3  (Repucci and Slipowitz 2020) For example, the news came from China about a ban on entering Buddhist temples for acts of worship while they were open for tourism; public health regulations served to restrict or persecute religious minorities; mosques were singled out in India as places of dangerous contagion; stricter quarantines were imposed on some citizens because of their religious affiliation. Many governments have used public health control regulations as political repressive and discriminatory measures against religious groups.

4   With the intention of allowing spiritual activity to continue, the WHO published an interim guide in April 2020: "Practical considerations and recommendations for religious leaders and faith-based communities in the context of COVID-19". This document acknowledges that religious leaders, faith-based organizations and faith communities can provide pastoral and spiritual support during public health emergencies and other health challenges and can advocate for the needs of vulnerable populations. We can see specific accommodation recommendations in this document—for example, recommendations to hold gathering outdoors or with fewer people, suggesting multiple services with a few attendees—among many other reasonable and effective suggestions to maintain religious activities during the pandemic. Unfortunately, many countries decided not to follow this guideline with the already-known restrictive consequences for religious freedom.

5   For example, in Spain, in October, the executive established a new protocol to rationalize the exercise of freedom of worship and the protection of health: "The permanence of people in places of worship is limited by fixing, by the authority competent corresponding delegate, of capacity for religious meetings, celebrations and gatherings, considering the risk of transmission that could result from collective meetings. Said limitation may not affect in any case the private and individual exercise of religious freedom", there recognizing the importance of not restricting the exercise of this individual right. In this way, the legal limbo in which the acts of worship had remained in due to the March regulation was corrected.

6   "Access to the judicial system remained open so that citizens and legal entities could challenge, in court, the provisions and implementing acts. This access to the legal system, I argued, as a basic pillar of Rule of Law, revealed that democracy kept functioning during the pandemic" (Von Münchow 2020).

7   This is the main reason to include the rule of law as a general principle. Even if the Venice Commission reached the conclusion that Rule of Law was indefinable, a core elements checklist must be used as an operational tool to control state's compliance with the rule of law: legal certainty, prevention of abuse, equality before the law and access to justice. This checklist also addresses specific, topical challenges to the rule of law: corruption and conflicts of interest, collection of data and surveillance (CoE 2016).

8   As will be discussed, because freedom of thought, conscience and religion undoubtedly play a key role in the formation of ideological pluralism, ignoring the guarantee of protection of this right is probably the most direct way to weaken the democratic quality of a society.

9   Secularism involves a complex requirement, which can be classified according to Taylor into the categories of the trinity of the French Revolution: no one must be forced in the domain of religion; there must be equality between people of different faiths or basic beliefs; no religious entity can enjoy a privileged status; and finally, all spiritual families must be heard, included in the ongoing process of determining what society is about and how it is going to realize these goals (Taylor 2010).

10  To better understand the role that government policy regarding religious freedom plays in strengthening democracy (Gill 1999).

11  For the purpose of this research, the collective nature of religion, organized as a "church", must be considered a fundamental requirement for the approach presented here. For the collective approach to religion, see (Durkheim 1995).

12  The institutional approach must be understood according to Amartya Sen's theory of institutions (Sen 2009).

13  We agree with Durham's point of view on this topic (Durham 2004).

14  This is a particularly valuable category within the United Nations model, since for a successful solution to the sustainable development goals, they must be based on collegiate solutions in which both political power, private structures and organizations representing civil society participate (Tomalin et al. 2018).

15  "Governments can leverage the trust, reach and practical support of religious leaders to deliver effective public-health responses, because where confidence in and reach of government is fragile, trusted interlocutors are vital to the success of public-health responses. In fact, religious leaders can support behavioural change and public health messaging and provide facilities and community services". https://institute.global/sites/default/files/inline-files/Tony%20Blair%20Institute%2C%20Working%20With%20Religious%20Leaders%20to%20Support%20Public%20Health%20Measures.pdf, accessed on 11 April 2021.

16  Religious freedom is a complex right, not only because of the varied manifestations of worship but also because of the high degree of subjectivity granted to determine when a personal behavior becomes a religious expression. Such a question necessarily leads to considering the degree of sincerity of those beliefs; however, to answer this question, we must address the collective dimension of religious freedom as a fundamental right. A religiosity test can be the degree to which an individual should be entitled to take responsibility for his convictions—a situation that leads to the extreme affirmation of denying the possibility of an effective protection of this right (Sullivan 2018).

17    The Universal Declaration and the norms of the Inter-American System for the protection of human rights are taken as a reference because the regional focus of this work is centered on the Peruvian and Latin American systems.

18    "The Universal Declaration of Human Rights (UDHR) does not define the terms 'thought', 'conscience' and 'religion'" (Scheinin 2000).

19    "[w]hat has proven to be one of the most influential statements of religious rights of humankind yet devised entered into the international arena with no further light shed upon its meaning." (Evans 1997).

20    Thus, converting freedom religious into a freedom that is reduced to the scope of the expression or manifestation of acts of worship.

21    With significant controversy among Islamic countries, as can be seen in the negotiation phase of the treaty (Clarke 1993).

22    The best way to guarantee the effective protection of freedom of thought is through non-coercion, allowing ideas to be freely expressed without censorship practices and with no limits on freedom of information other than those necessary for the protection of the fundamental rights of third parties.

23    When the fulfilment of a mandate of our conscience is placed before the fulfilment of a legal duty, the question of conscientious objection arises. Although it is true that the conscience enjoys absolute protection in the forum internum, and no one can force us to act against it, a different question concerns the responsibility produced by that action of our conscience leading it to disregard a legal mandate in the forum externum. It will then be necessary to verify the degree of involvement of one's conscience with this moral mandate; it will be necessary to determine the eventual damage to other legal rights and rights; and it will be necessary to confirm the degree of seriousness or truth that the act contains (Prieto Sanchís 2006).

24    The ECHR has made it very clear in Leyla Sahim v. Turkey, recalling that: "as protected by Article 9 (of the ECHR), freedom of thought, conscience and religion represents one of the foundations of a democratic society within the meaning of the Convention. This freedom figures in its religious dimension among the most essential elements of the identity of believers and their conception of life, but it is also a precious asset for atheists, agnostics, sceptics or the indifferent. It is about pluralism,—achieved in a very expensive way over the centuries—that could not be dissociated from such a society".

25    Particularly the International Covenant on Civil and Political Rights (ICCPR), which acknowledges that states may need additional powers to address exceptional situations. Nevertheless, it is well established that "No state party shall, even in time of emergency threatening the life of the nation, derogate from the Covenant's guarantees ( … ) and freedom of thought, conscience and religion. These rights are not derogable under any conditions even for the asserted purpose of preserving the life of the nation" (United Nations 1985).

26    Article 4, paragraph 2 of the Covenant explicitly prescribes that no derogation from the following articles may be made to Article 18 (freedom of thought, conscience and religion); however, the notifications that Peru submitted in compliance with this obligation have always been generic: "The Secretary-General received from the Government of Peru a notification dated 19 March 2020, made under article 4 (3) of the above Covenant, regarding the declaration of a state of national emergency for a period of fifteen (15) calendar days by Supreme Decrees No. 044-2020-PCM of 15 March 2020, No. 045-2020-PCM of 17 March 2020 and No. 046-2020-PCM of 18 March 2020". Further extensions to the state of national emergency during pandemic were reported every time they were declared, but always in a general and unspecific document. On the other hand, Peru is a signatory state of the Pact that has accumulated the highest number of notifications (244 until January 2021) of a state of emergency, essentially because it does not have a constitutional regulation regarding the state of emergency.

27    In order that the Committee can perform its task to monitor these emergency laws, states should include in their reports sufficient and precise information about their laws and practice in the field of emergency power. Unfortunately, the information they send is usually too generic and inadequate for the Committee to properly carry out this control work. A fundamental requirement for any measures derogating from the Covenant, as set forth in Article 4, paragraph 1, is that such measures are limited to the extent strictly required by the exigencies of the situation. This requirement relates to the duration, geographical coverage and material scope of the state of emergency and any measures of derogation resorted to because of the emergency.

28    At that moment, IACHR had already observed that different states in the region responded to exponential increases in the number of infections by declaring states of emergency, states of exception and states of disaster on the grounds of so-called public calamity or health emergencies through presidential decrees and various types of regulations to protect public health; many of then officially informed the OAS that they had suspended guarantees as per Article 27 of the American Convention (OAS 2020).

29    In this respect, the context of the case of the Pentecostal church that reached the Supreme Court of the United States in April 2020 must be understood. The discussion here concerned this religious group: given that they carry out the sacramental elements of their worship in collective headquarters, prohibiting religious assembly implies that the right to religious freedom of this community will be left without content. In this first case, the Supreme Court did not rule in favor of the religious community; it gave the legislator freedom to maintain the restrictions, but it intuited (and later recognized in November 2020) that there was actually an unjustified violation of religious freedom.

30    Particularly the layers of content that are in the forum internum.

31    Norms on Canon Law are clear in establishing that the sacraments require a physical presence of the priest to be imparted. "The Internet is relevant to many activities and programs of the Church— evangelization, including both re-evangelization and new evangelization and the traditional missionary work *ad gentes*, catechesis and other kinds of education, news and information, apologetics, governance and administration, and some forms of pastoral counselling and spiritual direction. (…) Virtual reality is no substitute for the Real Presence of Christ in the Eucharist, the sacramental reality of the other sacraments, and shared worship in a flesh-and-blood human community. There are no sacraments on the Internet; and even the religious experiences possible there by the grace of God are insufficient apart from real-world interaction with other persons of faith" (Pontifical Council for Social Communications 2002).

32    For more details on the role of neutrality in models of church–state relationships, see (Ruiz Miguel and Miranda 2014).

33    For further reliable information to compare church and state models, see (The Religion and State Project, https: //www.thearda.com/ras/about/, accessed on 11 April 2021) (RAS 2015).

34    Halmai concludes in his study that "the worldwide resurgence of religion forces liberal constitutionalism to adapt religious rights to different state–church relationships" (Halmai 2017).

35    In this context, it is possible to consider what future effect these events will have on people's level of religiosity. Will the secularization of society increase or will there be a resurgence in human beings' approach to divinity? Certainly, this is an important question that unfortunately cannot be resolved at this time or in this work; however, it can be intuited that the political model and the decisions it adopts within a religious policy strategy will have an enormous impact on the future evolution of religiosity.

36    Tutela 1a Inst: 2020-4398 Accionado: Presidencia de la República y otros. Accionante: Edna del Carmen Benítez Casanova.

37    For opinion from Chile, see (Celis 2020; Patiño 2020; Navarro Floria 2020).

38    This is the conclusion of Professor Saldaña analyzing the Mexican case (Saldaña Serrano 2020; Navarro Floria 2020).

39    The ACHR departs from the triad of "thought, conscience and religion" that appears in the main human rights treaties and in many of the constitutional texts of the second half of the 20th century. It appears that it decided to separate religious freedom from freedom of thought with the intention of linking freedom of expression with freedom of ideas from which every democratic state is nourished. The Inter-American Court of Human Rights has given reasons in several of its judgments to understand the special treatment that freedom of thought maintains with freedom of expression within the IAHRS, relating it to the importance of freedom of the press as a guarantee of the democratic quality of states (Mosquera 2017).

40    Those could be described as "weak establishment" church–state systems.

41    However, in practice almost all Latin American countries fit the conclusions reached by Fox in his study on separation and secularism. He summarises that: "most state, even those which declare in their constitutions that they are secular or follow a separationist policy, do not follow these policies" (Fox 2011).

42    Essentially to those other than the Catholic Church since its recognition is obtained through its legal personality as an international subject, and is expressed in the agreements of a concordant nature that remain in force.

43    Agreements equivalent to those that the Catholic Church has that allow the effective realization of the scope of cooperation between institutions.

44    For more detail on the effects of regulations on religious freedom, see (North and Gwin 2004).

45    Article 80A of Supreme Decree No. 019-2001-JUS, which approves the regulation of organization and functions of the Ministry of Justice, published on 20 June 2001, later modified by Article 2 of Supreme Decree No. 026-2002-JUS, of 26 July 2002.

46    This registry will be published in 2003 with the approval of Supreme Decree No. 003-2003-JUS, and it was regulated that same year (Ministerial Resolution No. 377-2003-JUS (13 October 2003) that implements the Registry of Confessions other than Catholic (RCDC) and approves its Applicable Norms). It began its task of registering the different denominations that religious groups adopt in the Peruvian territory with remarkable success if we verify its statistics since it gave entry to 142 religious entities in the 7 years that it was in operation.

47    Only after verifying that the procedure was poorly planned did he choose to review it through a series of changes incorporated into the law on religious freedom (Mosquera 2019).

48 Legal actions in the United States, Germany, Italy and France were filed by religious authorities against the state authorities that had established the restrictive measures. The lawsuits mentioned in Colombia or Argentina were filed by private citizens. No Roman Catholic Diocese of Brooklyn v. Cuomo could be found at the judiciary in South America.

49 "Whilst 'states of emergency' or similar exceptional regimes may allow for a more rapid, flexible and effective response, they limit the application of normal checks and balances" (CoE 2020).

50 Especially in the initial phase where the communities made the decision to adapt to the virtual format in order to attend to their faithful and worrying above all about giving support to their community.

51 Adapting liturgical celebrations and funeral rites, among others. In this second state, there were also religious communities that opted for a position of denial of the facts in a kind of confrontation between science and religion.

52 Taking legal action and claiming against restrictions and proposals by the government.

53 As we can see in the Colombian regulation: "once the conditions surrounding the activities of the religious sector have been analysed, and in accordance with the information provided by the National Board of Social Action of the Ministry of the Interior and the participation of leaders of the different confessions and religious communities of the country, this Portfolio prepared the special biosafety protocol to be applied in this sector, which is adopted by means of this resolution". Resolution No. 1120, 3 July 2020.

54 Protocol for the religious activities of the Catholic Church in times of pandemic. Peruvian Episcopal Conference. "In all this time of national and health emergency, we have complied with and supported the measures established by the Government to prevent the spread of the COVID-19 pandemic. These measures obviously do not deny or impede the freedom to express our religious convictions or the worship that we need to offer to God. Once the state of national emergency (quarantine) has ended, each Bishop of the place will establish the date from which the faithful will be allowed to attend the temples for Eucharistic celebrations". 25 June 2020.

55 It is true that the circumstances of the Peruvian public health system also affect the decision, but they are not the only factor. This precarious situation of the public health system in the specific Peruvian case is not a novelty; it had been specifically pointed out in timely reports prepared by the same government when presenting a report to the OECD in 2017, and in many others presented to international institutions (CEPAL 2020).

56 Proyecto de ley, No. 6391/2020-CR, proyecto de ley que dispone la reapertura de templos de toda confesión o denominación religiosa, sean parroquias, iglesias y similares; cumpliendo con los protocolos de bioseguridad, en el marco de la declaratoria de emergencia nacional a causa del COVID-19, 8 octubre 2020.

57 Decreto Supremo que modifica el artículo 5 del Decreto Supremo N° 170-2020-PCM. Artículo 5.—De la apertura de los templos o centros de culto religioso. "Se autoriza a partir del lunes 02 de noviembre de 2020, a que las entidades religiosas abran sus templos y lugares de culto para recibir a sus miembros, fieles y público en general, para la profesión individual de su fe, con un aforo no mayor a un tercio (1/3) de su capacidad total, y excepcionalmente podrán celebrar sacramentos y ceremonias especiales afines según su culto, debiendo adoptar y cumplir las normas sanitarias emitidas por la Autoridad Sanitaria Nacional y las medidas aplicables del Estado de Emergencia Nacional. A partir del 15 de noviembre de 2020, las entidades religiosas podrán celebrar ritos y prácticas religiosas de naturaleza colectiva, con un aforo no mayor a un tercio (1/3) de la capacidad total de los templos o lugares de culto, según los protocolos debidamente acordados por la Autoridad Sanitaria Nacional y en concordancia con las medidas del Estado de Emergencia Nacional". In the same context, the Health Department aproved "Directiva sanitaria N° 121-MINSA/2020/DGIESP, Directiva sanitaria que establece medidas para el reinicio de actividades religiosas o de culto en el marco de la emergencia sanitaria de la COVID-19", 30 October 2020.

58 COVID-19: Italy returns to strict lockdown for Easter. https://www.bbc.com/news/world-europe-56621342, accessed on 11 April 2021.

59 "The COVID-19 pandemic is harming health, social and material well-being of children worldwide, with the poorest children, including homeless children and children in detention, hit hardest. School closures, social distancing and confinement increase the risk of poor nutrition among children, their exposure to domestic violence, increase their anxiety and stress, and reduce access to vital family and care services. Widespread digitalisation mitigates the education loss caused by school-closures, but the poorest children are least likely to live in good home-learning environments with internet connection" (OECD 2020b).

60 The impact on funeral services has been significant. In this specific manifestation of the posthumous act of worship of the faithful, including accompanying and mourning the death of a loved one, the strict restrictions imposed during this pandemic have been an effective violation of three dimensions of the health of the relatives who have buried a loved one in these conditions. The WHO has reminded us that health deserves a complete definition; it is not only physical confinement and restrictive measures on rights that have put health at risk in a multilevel sphere.

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
