# Peer review of "The Impact of the Church–State Model for an Effective Guarantee of Religious Freedom: A Study of the Peruvian Experience during the COVID-19 Pandemic"

_laws, 2021_

Round 1

Reviewer 1 Report

The article aims to investigate the way different nations, and in particular Peru, deals with rleigious freedom during the covid-19 pandemic. It has been argued that the right to religious freedom is better guaranteed in nations where religious entities are recongized by the state then in nations where this is not the case. This is an interesting point, but the main problem with this argument is that it is too generalistic. For instance, religions are not legally recognized or institutionalized by the Dutch state, but the right to practice one's religion seems to be far more protected in the Netherlands than in its neighbouring country Belgium, where there is a legal recognition of different religions. In spite of this legal recognition, Belgian religious groups are, compared to religious groups in the Netherlands, far more restricted in their exercise of religious freedom during covid-19 (which even led to a constitutional court case). This is but one example which shows that the author's argument is not convincing - at least not in its generalized form.

In addition, I think the author wants to say too much in one article. In my opinion, it would be better to compare only 2 or 3 nations (e.g. Peru and 2 other nations, for instance with a different church-state model) more in debth, than to give a kind of general overview and than look at the concrete case of Peru. By doing this, the author could delve more into the details of the different countries and overhastly generalized conclusions could be avoided.

Besides, some statements are taken for granted, without nuance. Also here, improvement is needed. A few examples:

p. 3: "So, if science did not have full surety regarding how to deal with the disease, how could the measures for prevention and control be placed without causing restriction on fundamental rights?" I would say that this is, unfortunately, how science works. Criticizing covid-19 restrictions because they are embedded in scientific hypothesis is a bit strange, becuase this is just how it works. The cost of not taking restrictive measures would be too high - even if some fundamental rights are temporarily restricted (by the way: not only the right to religious freedom; also other fundamental rights are restricted - it is always a matter of proportionality (which is not that often mentioned by the author).

p.5 line 116: it is taken for granted that gathering for gym is different from gathering for religious ceremonies. But why is this so? There is certainly a lot of philosophical literature about this topic (inspiration can e.g. be found in  the Politics or religious freedom, edited by Sullivan et al.) and this seems to be ignored by the author. What makes religion so special that it needs different treatment in pandemic situations? Just claiming that religion is different from gym is, according to my opinion, not sufficient. Moreover, claiming that "The truth is that faith is an essential element of human nautre" (p. 7 line 161) is neither convincing: culture can for instance also be seen as 'an essential element of human nature' and so is family life, the need to embrace people, etc. These things are also restricted in order to guarantee peple's health and safety...

p.  the author seems to favor a regime wherein religions are legally institutionalized, as this would lead to more (individual) religious freedom. However, (S)he seems to ignore the important work of e.g. Sullivan (The Impossibility of Religious Freedom) and Hurd (Beyond Religious Freedom), wherein this practice of institutionalizing religion is criticized, a.o. because it leads to an unequal treatment of lived religions and smaller - non-institutionalized - religious practices. 

What also strikes me is that art.9 of the EHRM is not mentioned. In an article wherein different policies towards religion in covid-19 are compared, this is at least strange, especially because the second paragraph of this article allows the restriction of religious freedom in order to protect public health and national security. It owuld be good to insert this somewhere in the text (maybe in the introduction?)

p. 21 line 581 ff is incorrect (because too generalistic): not everywhere in the world did churches and temples reopen after November 2020. In Belgium for instance, the covid statistics were quite optimistic, but churches were (different from e.g. non-essential shops and museums) not allowed to reopen their door. After a court case, the rules for churches were adapted, but they were still quite rigid for churhes and other religious communities. 

p.22 footnote 68: I wonder why Belgium and Switzerland have, according to the author, "shown very good practice in the application of cooperation measures such as dialogue and harmonization." This should be explained.

p.23, line 616 ff: religion is, from an holistic perspective, seen as part of the physical integrity. But doesn't the same go for sports, wellness centra, yoga centra, etc? I do not have a problem with this holistic view, but one should be honest here and admit that this does not only count for religion, but also for many other things, which contribute to the overall well-being of people...

p. 25, line 677: "Religion deserves this exception because its activity is the essence of human right". I don't get this and I don't agree. This is just postponed, without any clear legitimation; see also line 685 ff.

p.26 line 730: this is not correct (too general - see what I mentioned about eg Belgium and the Netherlands)

I am aware that the author cannot include everything in one article. Therefore I would recommend a narrower, comparative focus, which will enable the author to go more in depth and add more nuances. Of course, the author should not agree with all my comments and I am aware that you cannot know everything about everything. But if one wants do draw generalistic conclusions, some 'taken for granted' points need more ground and clarification. I therefore recommend publicaiton after major revision;

Author Response

Truly appreciating the comments.

The analysis will be focus on the Peruvian and Latin American context. Some of the general conclusions will be removed or re-drafted at the work. The paper tries to offer arguments that strengthen the models of church-state collaboration. Proper clarification will be made in the text. The paper will take a narrower focus.

Thank you.

Reviewer 2 Report

The ms tackles an important and pressing topic, namely the effect of religion-state relations on religious freedom in the context of a public health emergency. The author demonstrates a strong grasp of the empirical issue, particularly in terms of the Peruvian case, as well as its normative implications. Much of the theoretical framework for understanding religion state relations, such as the "layers" of religious freedom and the relationship between its individual and collective dimensions, are compelling and well-argued. 

The main weaknesses of the piece are its heavy normative framing and its insufficient engagement with the growing body of literature on empirical varieties of religion-state relations.

In terms of the first, the ms makes a number of assertions that are grounded in a specific, normative stance that appears highly suspicious of public health measures and state action. These undermine its credibility, making it read like an editorial or a partisan piece rather than a research piece, and are ultimately unnecessary for its fundamental argument. At times, the author seems to doubt that physical distancing is a necessary component of public health policy in the covid pandemic (e.g. pp 3-4) and that the state is acting maliciously or with reckless disregard for religious groups (e.g. pp 4-5). Later in the ms, the author adopts a more reasonable stance that notes the varieties of possible responses and compares the "total lockdown" approach in Peru to the more graduated policies adopted by other states - this is a much more useful position, particularly if it were more systematically documented. Could the author create a typology of restrictions and attempt to document which countries fall where?

This leads to the second problem, namely that patters of religion-state relations are described with anecdotes and/or with a very broad brush, and with seeming unawareness of the growing body of works on the varieties of religion state relations. For example, see the works of DT Buckley (2015, 2017), MD Driessen (2010, 2014), LF Mantilla (2016, 2019) and especially J Fox (2008, 2015). Engagement with this literature would significantly strengthen the argument, and allow the author to describe patterns of religion-state relations in ways that make it less vulnerable to the critique that he/she is cherry-picking examples. 

Importantly, the author seems to set aside the problem of state capacity. That the policy response of European countries and the United States has been generally better (more measured, less draconian) than that of (many? all?) South American countries does not seem like much of a riddle, nor does it require a dissection of religion-state relations to understand. The former have more money, expertise, and bureaucratic capacity (and less corruption and political dysfunction). Peru has had three unelected presidents in a row due to its never-ending political crises. Perhaps comparing the inadequate Peruvian response to a "better" response by another South American country - or another comparable country with a different religion-state pattern - would be more compelling. 

Finally, the English writing, while legible, requires substantial revision before the ms would be eligible for publication. 

Author Response

We truly appreciate all the constructive comments and suggestions from the reviewer and try to adopt all the suggestions in our revised manuscript. We will rewrite many parts of the revised manuscript in response to the reviewer’s comments and we will try to include the literature mentioned by the reviewer. Although we must specify that this work is based on the models used by the legal literature on the church-state relationship. 

Round 2

Reviewer 1 Report

Compared to the previous version, this article has clearly improved: the scope is less wide (focus on international HR legislation and then on Peru), which contributes to the overall analysis of the article. Besides, some relevant sources are added and the argumentation is more structured. One of my main concerns is that the article needs profound language editing before it can be published, as there are numerous grammatical and other mistakes (too much to sum up here or to highlight in the text).

Besides, a few other remarks:

  • p.5 line 94 refers to France and America and then, lin 98 refers only to European states (and thus not to the US). This is not consequent.
  • p.5 line 107: 'religious dogma' is considered to be an essential part of 'religion'. This view is, however highly contested among religious studies scholars and sociologists (which emphasize e.g. the concepts of lived religion; multiple belonging; bricolage religieux; belonging without believing...) and is indebted to a monotheïstic, western concept of religion. This needs nuance (or this should at least be mentioned). Asthe overall argument is also based on the idea of religious dogma (see e.g. p. 8 line 170 ff), the argument also needs fine-tuning in this regard.
  • p. 19 line 453: the truth is... whose truth?
  • p.24: conclusions: this is now just a summing up of some concluding remarks., which is good for a ppt presentation, but not for a conclusion of an academic text.  This part should therefore be written in a fluid text.
  • Unfortunately, I did not have enough time to check all the references, but I propose that the authors profoundly check this and that a (language) editor can also do this job. 

Author Response

Response to Reviewer 1 Comments

Compared to the previous version, this article has clearly improved: the scope is less wide (focus on international HR legislation and then on Peru), which contributes to the overall analysis of the article. Besides, some relevant sources are added and the argumentation is more structured. One of my main concerns is that the article needs profound language editing before it can be published, as there are numerous grammatical and other mistakes (too much to sum up here or to highlight in the text).

Response: I appreciate the time and effort that both reviewers have dedicated to providing such valuable feedback on my manuscript. I sincerely appreciate the recommendations and suggestions. The style review is indeed important to ensure that the document reaches the necessary quality. The review has been a positive experience that has allowed me to improve my work. I appreciate the time and help provided by the reviewers. Thank you.

Besides, a few other remarks:

  • p.5 line 94 refers to France and America and then, lin 98 refers only to European states (and thus not to the US). This is not consequent.

Response: In fact, it is a consistency error. It will be modified.

  • p.5 line 107: 'religious dogma' is considered to be an essential part of 'religion'. This view is, however highly contested among religious studies scholars and sociologists (which emphasize e.g. the concepts of lived religion; multiple belonging; bricolage religieux; belonging without believing...) and is indebted to a monotheïstic, western concept of religion. This needs nuance (or this should at least be mentioned). Asthe overall argument is also based on the idea of religious dogma (see e.g. p. 8 line 170 ff), the argument also needs fine-tuning in this regard.

Response: It is a translation error. What is meant is religious teaching. Proper clarification will be made; following the reviewer's suggestion.

  • p. 19 line 453: the truth is... whose truth?

Response: It is a linguistic resource. It will be replaced by another expression.

  • p.24: conclusions: this is now just a summing up of some concluding remarks., which is good for a ppt presentation, but not for a conclusion of an academic text.  This part should therefore be written in a fluid text.

Response: Right. They will be rewritten in paragraph format.

  • Unfortunately, I did not have enough time to check all the references, but I propose that the authors profoundly check this and that a (language) editor can also do this job. 

Response: In addition to the above comments, all spelling and grammatical errors pointed out by the reviewers will be corrected.

Reviewer 2 Report

The authors have made important improvements, most notably in terms of their engagement with the literature. While they opted not to pursue some of the suggestions in my previous review (e.g. addressing differences in state capacity) and did others only slightly (especially toning down the normative tone) this now seems like a deliberate choice of emphasis resulting from their focus on legal/normative frameworks. I leave these matters, particularly in terms of normative tone, to the editors' discretion. There are still important and very necessary revisions to make in terms of English language use (e.g. replacing "The Latin American singularity" with "Latin American distinctiveness" or something like that). 

Author Response

Response to Reviewer 2 Comments

The authors have made important improvements, most notably in terms of their engagement with the literature. While they opted not to pursue some of the suggestions in my previous review (e.g. addressing differences in state capacity) and did others only slightly (especially toning down the normative tone) this now seems like a deliberate choice of emphasis resulting from their focus on legal/normative frameworks. I leave these matters, particularly in terms of normative tone, to the editors' discretion. There are still important and very necessary revisions to make in terms of English language use (e.g. replacing "The Latin American singularity" with "Latin American distinctiveness" or something like that). 

Response: I appreciate the time and effort that both reviewers have dedicated to providing such valuable feedback on my manuscript. I sincerely appreciate the recommendations and suggestions. The suggestions that have not been included, will nevertheless be taken into consideration for the future development of this research.

Response: The style review is indeed important to ensure that the document reaches the necessary quality. The suggestion of the term ("Latin American distinctiveness”) made by the reviewer will be included.

The review has been a positive experience that has allowed me to improve my work. I appreciate the time and help provided by the reviewers. Thank you.
